# Collapse and continuity: A multi-proxy reconstruction of settlement organization and population trajectories in the Northern Fertile Crescent during the 4.2kya Rapid Climate Change event

**Dan Lawrence** [1]*, **Alessio Palmisano** [2], **Michelle W. de Gruchy** [1]

**1** Department of Archaeology, Durham University, Durham, United Kingdom, **2** Department of Ancient History, Ludwig-Maximilian University of Munich, München, Germany

* dan.lawrence@durham.ac.uk

**Data Availability Statement:** The minimal dataset is within the manuscript and its Supporting Information files. The digital archive related to this

## Abstract

The rise and fall of ancient societies have been attributed to rapid climate change events. One of the most discussed of these is the 4.2kya event, a period of increased aridity and cooling posited as the cause of societal changes across the globe, including the collapse of the Akkadian Empire in Mesopotamia. Studies seeking to correlate social and climatic changes around the 4.2kya event have tended to focus either on highly localized analyses of specific sites or surveys or more synthetic overviews at pan-continental scales, and temporally on the event and its aftermath. Here we take an empirical approach at a large spatial scale to investigate trends in population and settlement organization across the entirety of Northern Fertile Crescent (Northern Mesopotamia and the Northern Levant) from 6,000 to 3,000 cal BP. We use Summed Probability Distributions of radiocarbon dates and data from eighteen archaeological surveys as proxies for population, and a dataset of all settlements over ten hectares in size as a proxy for the degree of urbanization. The goal is to examine the spatial and temporal impact of the 4.2kya event and to contextualize it within longer term patterns of settlement. We find that negative trends are visible during the event horizon in all three proxies. However, these occur against a long-term trend of increased population and urbanization supported through unsustainable overshoot and the exploitation of a drier zone with increased risk of crop failure. We argue that the 4.2kya event occurred during a period of unprecedented urban and rural growth which may have been unsustainable even without an exogenous climate forcing.

## Introduction

The rise and (especially) fall of ancient societies have been attributed to rapid climate change (RCC) events [1–4]. One of the most discussed of these is the 4.2kya event, a period of increased aridity and cooling initially posited by Weiss and colleagues as the cause of the

paper allows reproducible analyses and figures in the form of three scripts written in R statistical computing language. The digital archive is freely available at the following link via the repository Zenodo (http://doi.org/10.5281/zenodo.4322990).

**Funding:** This research was supported by the European Research Council under the European Union's Horizon 2020 research and innovation program (https://ec.europa.eu/programmes/horizon2020/en/h2020-section/european-research-council) for the project 'CLaSS – Climate, Landscape, Settlement and Society: Exploring Human Environment Interaction in the Ancient Near East' (grant number 802424, award holder: DL). The funders had no role in study design, data collection and analysis, decision to publish, or preparation of the manuscript.

**Competing interests:** The authors have declared that no competing interests exist.

collapse of the Akkadian Empire in Northern Mesopotamia at the end of the Early Bronze Age [5], and since implicated in societal changes across the globe [6,7]. The definitions of collapse put forward by proponents of the 4.2kya collapse hypothesis vary according to regional context, but generally include the widespread abandonment of settlements, taken as evidence for depopulation, and an overall simplification of social organization, in addition to the disintegration of political units. The hypothesis has generated a significant body of research, but there are few signs of a developing consensus on its validity. In this article we examine three different archaeological datasets between 6,000 and 3,000 cal BP, covering the period before, during and after the event, across the northern part of the Fertile Crescent. Site densities derived from individual archaeological surveys and summed probability distributions (SPD) of calibrated radiocarbon dates are used as proxies for population. The size and location of urban sites, defined as those over ten hectares in size, are used as a proxy for both population and social complexity. The goal is to investigate trends in population and settlement organization at a regional scale to assess the impact of the 4.2kya event.

Assessing the relationship between climate change and past human societies is fraught with difficulties, in part due to differences in the temporal and geographical extents of the datasets used [8,9]. Studies of the 4.2kya event generally derive past climate variability from proxy datasets including pollen, diatom, tephra and forminifera records and stable isotopes held in natural archives such as speleotherms [10]. Climate modelling has also begun to play a role in these debates [11,12]. All of these can be more or less well-constrained chronologically and are more or less susceptible to local conditions. As a result, although there is a degree of scientific consensus on the significance of the 4.2kya event, as evidenced by the recent decision by the International Commission on Stratigraphy to use it to demarcate the beginning of the Late Holocene subepoch [13], there is significant variability in its proposed duration and magnitude in local settings. In the Fertile Crescent, recent studies have tended to coalesce around a multi-centennial rather than -decadal duration between 4,300 and 3,900 cal BP, but estimations of the percentage change in rainfall range from 7% [12] to 50% [14], where they are specified at all. In this paper we do not address the debates in the climate literature but examine trends in data derived from the archaeological record before, during and after the period in which the various climate proxies indicate the event occurred. The archaeological record is subject to similar spatial and temporal difficulties. With some notable exceptions [15–17], studies seeking to correlate social and climatic changes around the 4.2kya event have therefore either focused on highly localized empirical analyses of specific sites or surveys, or taken a more synthetic overview at much larger, even pan-continental, scales [18].

Here we take an empirical approach at a large scale. Our study region encompasses the entire northern Fertile Crescent, an arc of plains and steppe which stretches from the Zagros Mountains in the east to the Taurus and Anti-Lebanon mountains in the west, situated today within Northern Iraq, South-Eastern Turkey, Syria, Lebanon and Israel-Palestine (Fig 1). We derive population estimates from raw counts of sites from 18 discrete archaeological surveys, SPDs of all published radiocarbon dates, and summed counts and areas of all known urban sites. Raw counts and settled areas of archaeological sites derived from landscape survey are the most common archaeological proxy for inferring past human population dynamics [19]. We use site counts rather than area for the survey data because several surveys available in the region do not include site size information, and because most sites are relatively small and stable in size [20,21]. Landscape surveys rely on material culture collected from surface assemblages, and especially pottery, for dating. Radiometric dating techniques provide useful boundary conditions for ceramic forms recovered at individual sites, but how far these can be projected regionally is unclear. As a result, the chronological precision available through landscape survey is limited. Over the past two decades, SPDs of archaeological radiocarbon dates

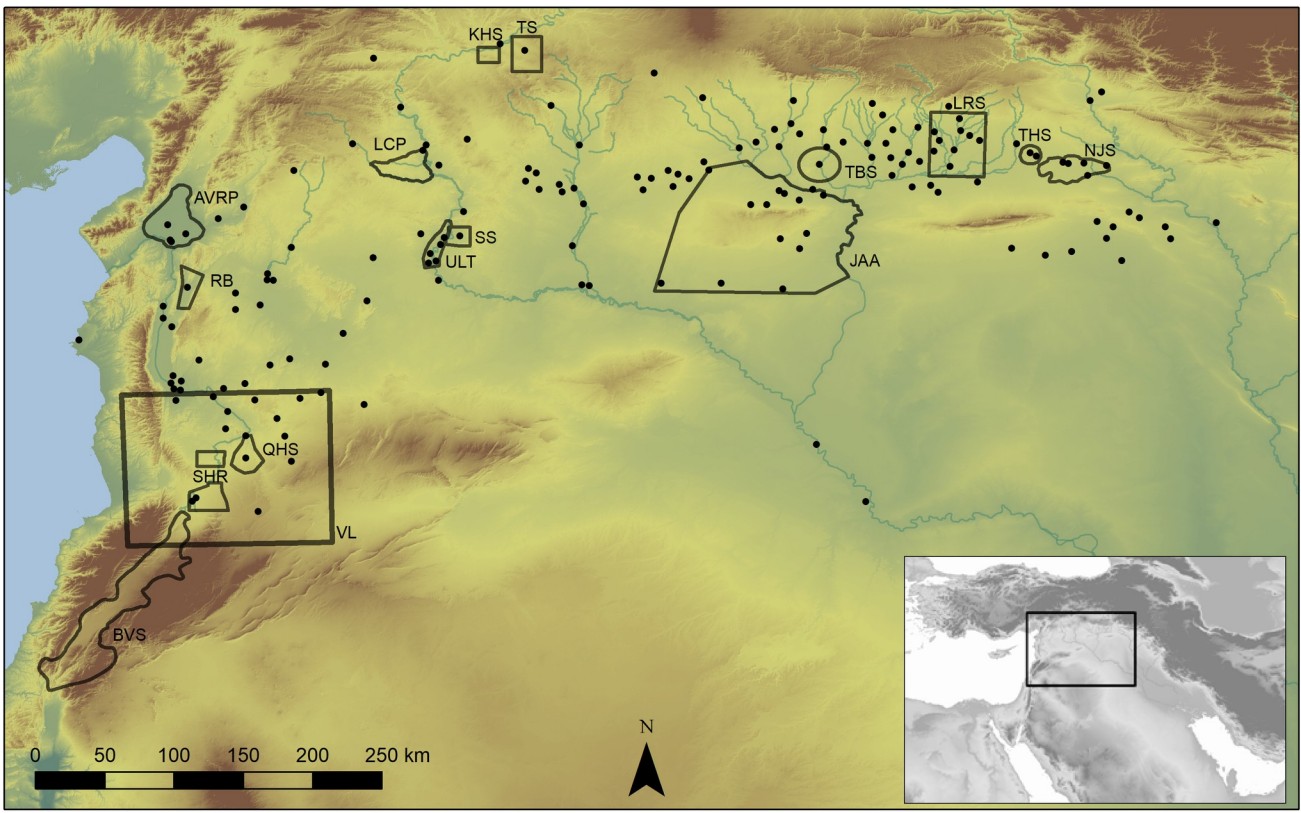

**Fig 1. Study area with urban sites and survey areas.** Black dots represent urban site points, polygons represent survey extents. See Table 1 for survey abbreviation. Background NASA JPL. NASA Shuttle Radar Topography Mission Global 3 arc second number. 2013, distributed by NASA EOSDIS Land Processes DAAC, https://doi.org/10.5067/MEaSUREs/SRTM/SRTMGL3N.003.

have begun to be used to model past demographic trends in prehistory, and these have the potential to document trends at higher levels of resolution. This latter approach has had limited impact in the Near East, although it is beginning to be used in earlier periods [22–26]. Both archaeological survey data and radiocarbon dates are subject to biases in research focus towards particular periods or regions, variations in the statistical methods adopted, taphonomic loss, the visibility of diagnostic artefacts, and research budgets [27–30]. Despite these issues, several studies focused on the Mediterranean basin have shown a good agreement in the demographic trends produced by the SPD of radiocarbon dates and archaeological survey data in the form of raw site count and estimated settlement size [31–36]. Here we use both proxies to model population through time.

Raw counts of settlement and SPDs do not capture estimated settlement sizes and therefore cannot provide information on settlement structure. In addition to these proxies we examine a dataset of all known sites which reach a size of ten hectares or more during the study period to investigate trends in urban population and organization. These sites are an order of magnitude larger than the average site size during this period [20] and, due to a strong bias in research agendas over the past hundred years, are more likely to have been excavated and systematically surveyed than small sites. This means changes in size are recognized and dated with greater precision. Larger sites demonstrate different trajectories to rural settlement over this period [36,37] and acted as both economic and political centers [38]. For example, evidence from both agricultural modelling [39,40] and combined textual and archaeological sources [41] indicate that by the second half of the Early Bronze Age (4,600–4,000 cal BP) large sites could

not farm sufficient land to feed their inhabitants and were reliant on rural production to supplement staple products. This implies systems capable of extracting surplus from rural sites, either through tribute and taxation or exchange. Defining urbanism in the past is contentious but most definitions emphasize concentration of population and a degree of social, political, and economic centrality relative to regional settlement [42,43]. These criteria are clearly met by the large settlements in the northern Fertile Crescent. This social centrality has also resulted in urban sites being used as a proxy for social complexity [44]. Urban decline or abandonment is central to narratives of civilizational collapse [45], including those of the 4.2kya event in our study region [46]. As such, in addition to providing a further dataset from which to model population trends, we use our urban dataset as a proxy for complexity of social organization.

## Materials and methods

### Archaeological data

The archaeological datasets (archaeological settlement data and radiocarbon dates) have been collected via harmonization of existing databases and publications to create three distinct georeferenced datasets (unprojected Lat Long coordinate system, WGS84 datum) for radiocarbon dates, survey site data and urban site data. The archaeological survey data have been collated from 16 published archaeological surveys (Table 1, see S1 Data for a full list of publications). These were selected on the basis of similarities in aims, methods and publication quality to ensure comparability in results. The survey dataset produced a total of 1157 sites divided into 2783 occupation phases (most sites were occupied in multiple periods) representing a reasonable picture of trends in the major agricultural basins of the study region [47]. The urban dataset (S1 Data) was compiled from survey and excavation reports supplemented by analysis of satellite imagery and builds on our previously published work [20]. This dataset includes 132 urban sites divided into 283 occupation phases. For the radiocarbon dataset we collated a total of 963 dates from 77 sites from published sources and existing digital archives [BANADORA:

**Table 1. Surveys used for site count data.**

| Survey Name | Abbreviation (Fig 1) | Number of Sites | Size (km2) |
|---|---|---|---|
| Amuq Valley Regional Project | AVRP | 178 | 535 |
| Bekaa Valley Survey | BVS | 229 | 2,749 |
| Jebel Abdul Aziz | JAA | 46 | 8,172 |
| Kurban Höyük Survey | KHS | 23 | 135 |
| Land of Carchemish Project Survey | LCP | 31 | 500 |
| Leilan Regional Survey | LRS | 201 | 1,462 |
| North Jazira Survey | NJS | 124 | 475 |
| Qatna Hinterland Survey | QHS | 22 | 305 |
| Rouj Basin | RB | 11 | 255 |
| Sites and settlement in the Homs Region | SHR | 118 | 400 |
| Sweyhat Survey | SS | 19 | 60 |
| Tell Beydar Survey | TBS | 57 | 316 |
| Tell Hamoukar Survey | THS | 42 | 125 |
| Titrish Höyük Survey | TS | 29 | 400 |
| Upper Lake Tabqa Survey | ULT | 14 | 400 |
| Vanishing Landscape Remote Survey | VL | 13 | n/a |
| **Total** | | **1,157** | **16,289** |

Note total area is not a direct reflection of total coverage due to overlapping survey areas.

48, CalPal: 49, CONTEXT: 50, 23, IRPA/KIK: 51, ORAU date lists, 52, PPND: 53, RADON: 54, TAY project: 55, see S1 Data for a full list of sources]. All of these radiocarbon dates come from archaeological contexts, with the majority being samples of bone, charcoal and seeds. Radiocarbon dates with poorly understood marine reservoir offsets (organic marine samples such as shells), a standard error greater than 300 years, or that do not have an anthropogenic origin (e.g. from environmental cores) have been excluded. The radiocarbon dates have been collected over a slightly broader time range spanning from 6,500 to 2,500 uncal BP in order to avoid edge effects. The total number of dates exceeds the suggested minimum threshold of 200–500 to produce reliable SPDs of calibrated radiocarbon dates with reduced statistical fluctuation for a time interval of 10–8,000 years [56,57].

## Inferring demographic trends from archaeological data

The archaeological settlement data derived proxies (raw site count, aggregated estimated settlement size) have been binned into a series of 100-year time slices starting at 6,000 cal BP (period $t_1$: 6,000–5,900 cal BP) and ending at 3,000 cal BP (period $t_{30}$: 3,100–3,000 cal BP). Individual sites and surveys use a variety of chronological schema with different degrees of chronological precision (different phase lengths). We applied aoristic approaches to deal with this temporal uncertainty [33,58,59]. The method assumes that the total probability of an archaeological event (site occupation phase in our case) within a given time span is 1, which indicates an absolute certainty that the site was in use in that time span. We then divide by the length of the site's chronological range to represent the probability of existence for each temporal block (implicitly adopting a default uniform assumption). For instance, using time-steps of 100 years, a Middle Bronze Age site-phase ranging from 4,000 to 3,600 cal BP has an aoristic weight of 0.25 for each time-step because the original 400-year time span is divided into four 100-year temporal blocks (4,000–3,900, 3,900–3,800, and so on). In addition, to mitigate the discrepancy between wide chronological uncertainties and narrower likely site durations, we applied Monte Carlo methods to generate randomized start of occupation periods for sites with low-resolution information, assuming a mean length of occupation of 100 years within each cultural period [33,59,60]. The resulting probabilistic distributions (aoristic weights, randomized start dates) provide a robust dataset to compare with other archaeo-demographic proxies.

The radiocarbon dates were processed using the R package *rcarbon* (version 1.4.1, Bevan and Crema 2020). We calibrated and summed the probability distribution of individual dates over a slightly broader time span from 6,500 to 2,500 uncal BP in order to avoid edge effects. We reduced the potential oversampling of specific site-phases by aggregating uncalibrated radiocarbon dates from the same site that are within 50 years of each other and dividing by the number of dates that fall in this bin [61]. This technique allows us to reduce biases introduced by oversampling of specific sites, chronological phases or events (for example, multiple dates being taken from a single floor layer). All other dates contribute to the curves at the resolution of the published date. A KDE map (bandwidth = 50 km) of the spatial distribution of radiocarbon dates shows uneven spatial sampling of radiocarbon dates due to research biases and different intensity of investigation (Fig 2A). Although almost half of the radiocarbon dates (n = 446, 46.8% of the total) are located in the Upper Khabur basin, aggregating uncalibrated radiocarbon dates within 50-year bins mitigate this issue. As a result, the number of bins drops to 29.4% of the total, reducing skew towards this region in the radiocarbon dataset to a similar level to site-phases of the settlement dataset (25.26%) (see Table 2).

The 963 radiocarbon dates have been grouped into 353 50-year bins (Fig 2B). Following the work by Weninger et al. [62] demonstrating that normalized calibrated dates of radiocarbon

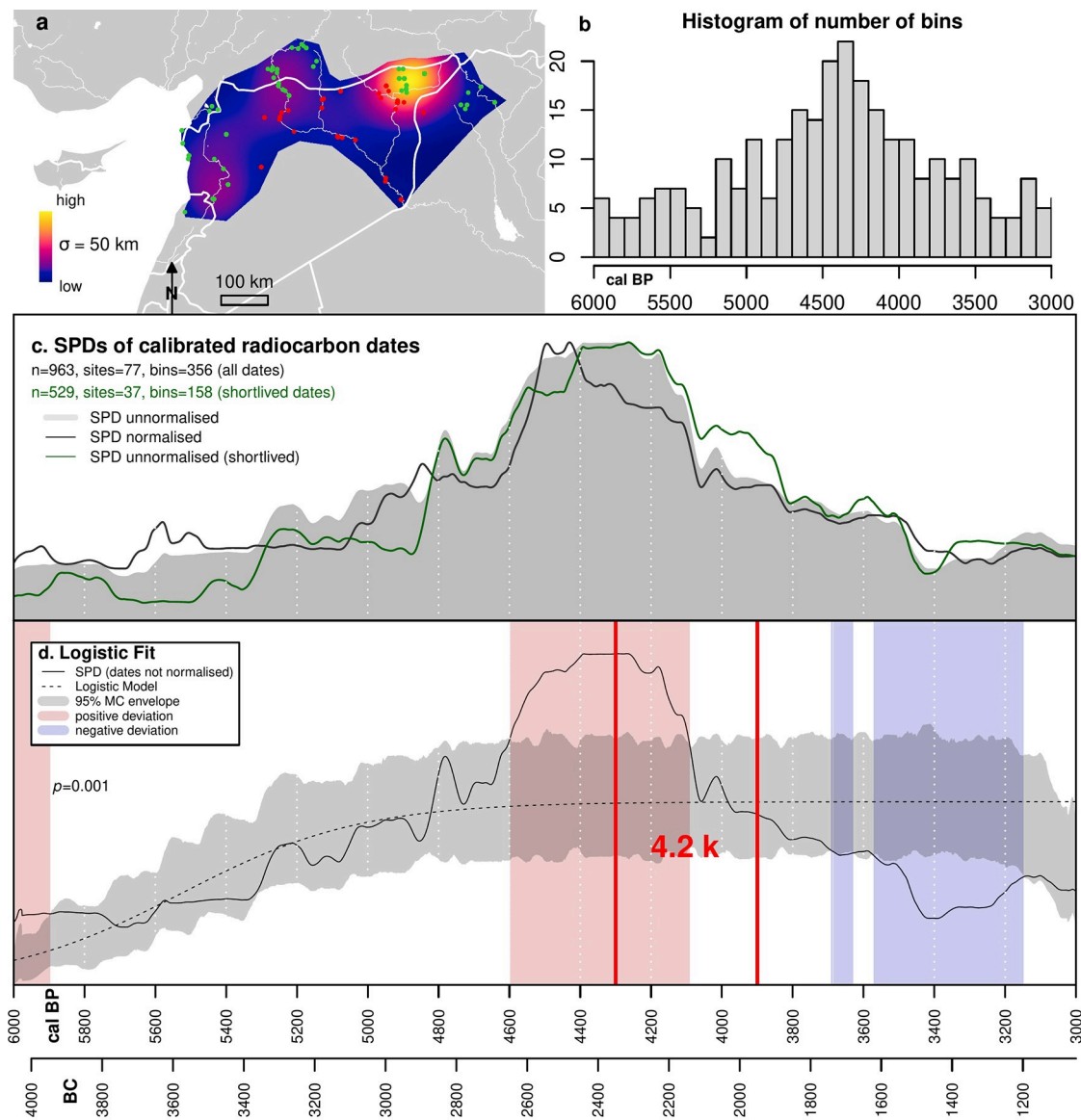

**Fig 2. Summed probability distribution dataset.** a: Kernel Density Estimate (50km bandwidth) of radiocarbon dates used. b. Histogram of bins and numbers of dates per bin. c: Summed Probability Distributions of calibrated radiocarbon dates. d: Summed Probability Distribution of dates superimposed on Logistic population model. Country outlines in map (a) from the GADM website (https://gadm.org/).

dates produces narrow artificial peaks due to the steepening portions of the calibration curve [we used IntCal20, [63]] we opted to sum unnormalized probability distributions [24,25,64]. We compared our observed unnormalized SPD of calibrated radiocarbon dates against a theoretical null model in order to quantitively test if the inferred demographic trends depict significant

**Table 2. Archaeo-demographic proxies in the Upper Khabur valley.**

| Upper Khabur Valley | Radiocarbon dates (n) | Bins (n) | Archaeological sites (n) | Site phases (n) |
|---|---|---|---|---|
| | **446** | **104** | **266** | **703** |
| % of the total | 46.8 | 29.4 | 28.08 | 25.26 |

patterns not derived by mere chance. We fitted a logistic growth model to the observed SPD and produced a 95% confidence envelope of 1,000 SPDs randomly generated to test statistically if the observed pattern differs from the null models [61,65,66]. Deviations above and below the 95% confidence limits of the envelope indicate respectively periods of population growth and decline greater than expected according to a logistic model of population growth. The null model assumes that the long-term rate of pre-industrial population growth fits a logistic curve in which population gradually increased before reaching a limit based on the carrying capacity of the land. Deviations above the model envelope therefore suggest carrying capacity is exceeded.

## Inferring spatial trends in archaeological data

In addition to overall decline, population movement is a common feature of collapse narratives [67], and in our region Weiss and colleagues suggest habitat tracking, whereby populations moved to areas with greater water resources as aridity increased, was one of the responses to the 4.2kya event [5]. In order to test this we have investigated demographic trends in all three datasets both as a whole and divided either side of the 300 mm rainfall isohyet. This value was chosen as it corresponds to the upper boundary of the so-called zone of uncertainty. In this zone, rainfed cereal agriculture is possible but relatively risky due to interannual rainfall fluctuations [37], and animal exploitation is focused on arid adapted species such as sheep and goat rather than cattle and pigs [68]. In the absence of adaptive strategies, we would expect aridification events to have more of an impact on communities living in this zone than in areas with higher rainfall. It should be noted that our isohyets are derived from modern data, and that the 300 mm isohyet line represents a long-term average rather than a fixed boundary. Examining datasets from either side of the boundary has proved useful in interpreting archaeological data of various kinds, including stable carbon isotopes [69], zooarchaeological finds [68] and site morphology [70]. We use it here as a useful heuristic for interpreting our datasets and would encourage future research to model our data using other thresholds. We also investigate changes in the spatial patterning using snapshot maps of both settlement datasets at 200-year intervals across our study period. These snapshots include all sites with an occupation phase which overlaps with the particular interval. 200-year blocks were chosen because this time length corresponds to the shortest phases in our dataset, ensuring changes are captured without repeating map series. Because the urban dataset is contiguous, it is amenable to kernel density estimate (KDE) analysis [71]. We use KDE weighted by site size to produce cluster-based density surfaces, using a bandwidth of 50km chosen to produce the clearest expression of results. The survey dataset is not contiguous because each survey is a bounded unit, and therefore KDE maps are not appropriate. Instead, we have mapped site counts expressed as densities, derived by dividing the total number of sites occupied within a given period by the total area ($km^2$) of the survey.

## Results

### Summed probability distribution

The normalized (black line) and unnormalized (grey shading) SPD of calibrated radiocarbon dates (Fig 2C) for the period spanning from 6,000 to 3,000 cal BP are very similar (pairwise Pearsons's coefficient r = 0.96, p-value < 0.01). Because most of our samples come from wood charcoal, we also generated an SPD of radiocarbon dates (in green) from only short-lived organic samples (e.g. bones, collagen, seeds). The SPDs including all dates and only short-lived dates (Fig 2C) are also highly correlated (r = 0.96, p-value <0.01), suggesting the old wood effect did not have a significant impact. Fig 2D shows a comparison between the observed SPD

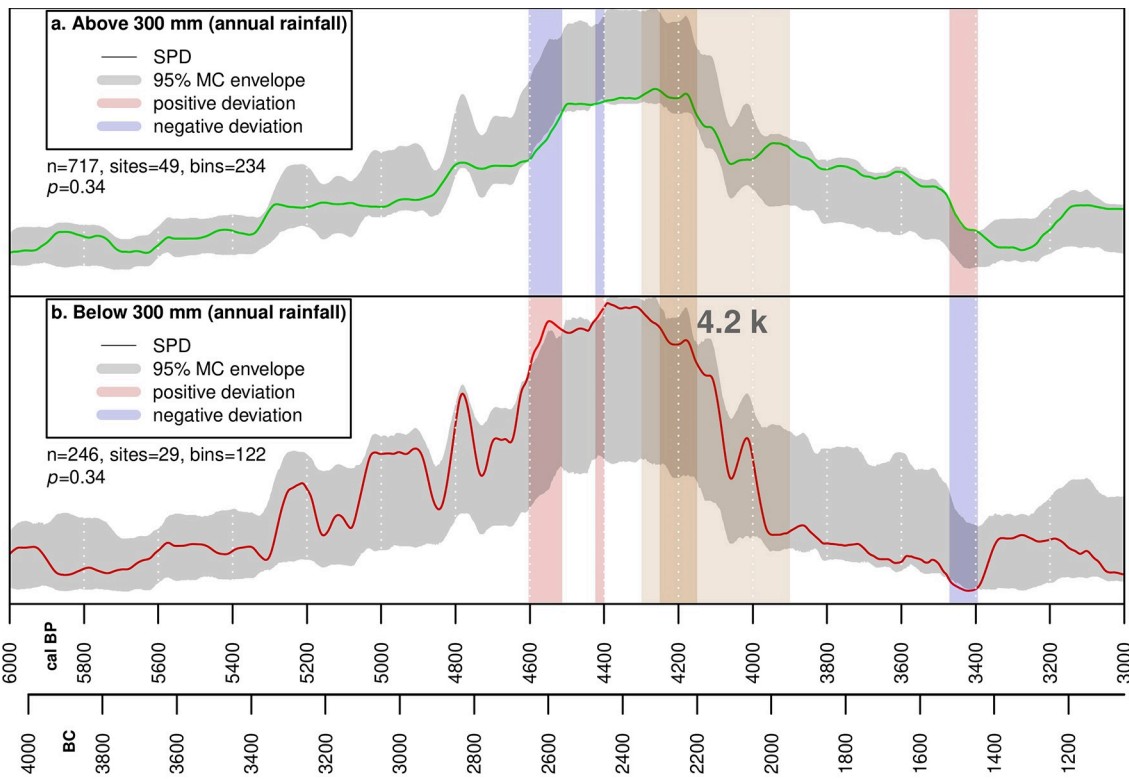

**Fig 3. Summed probability distribution permutation test.** a: Dates from sites located in areas with greater than 300 mm annual rainfall. b: Dates from sites located in areas with less than 300 mm annual rainfall.

of radiocarbon dates versus a logistic null model. The observed unnormalized SPD of radiocarbon dates (black solid line) differ significantly from the logistic null model as indicated by the global p-value (< 0.01). The curve suggests that population grew steadily from 6,000 cal BP and rapidly from around 4,700 cal BP before plateauing at around 4,500 cal BP and declining similarly rapidly from around 4,200 cal BP. The red shaded area indicates that population exceeded the expectations of the logistical model between 4,600 and 4,100 cal BP. Population continued to decline slowly, resulting in values below those predicted by the logistical model between 3,550 and 3,150 cal BP. However, these may be due to research biases in the radiocarbon dataset.

Fig 3 shows the SPDs subdivided above and below 300 mm. We have permuted subsets of all radiocarbon dates to determine whether the two curves significantly deviate from a 95% confidence envelope [72]. The demographic patterns in the two sub-regions do not differ significantly across the whole time-span (global p-value = 0.34), but between 4,600–4,500 cal BP and 4,450–4,400 cal BP the population located in the zone below 300 mm (Fig 3B) grows more than expected in comparison with population in the zone with higher annual precipitation (Fig 3B). The population in the annual rainfall zone below 300 mm shows a more marked decline during the drying phase between 4,300 and 3,900 cal BP but this remains within the modelled envelope.

## Settlement demographic proxies

The site-derived proxy indicators (raw count, aoristic weight, randomized start of date, estimated total area) and the SPD of radiocarbon dates have been normalized on a scale between 0

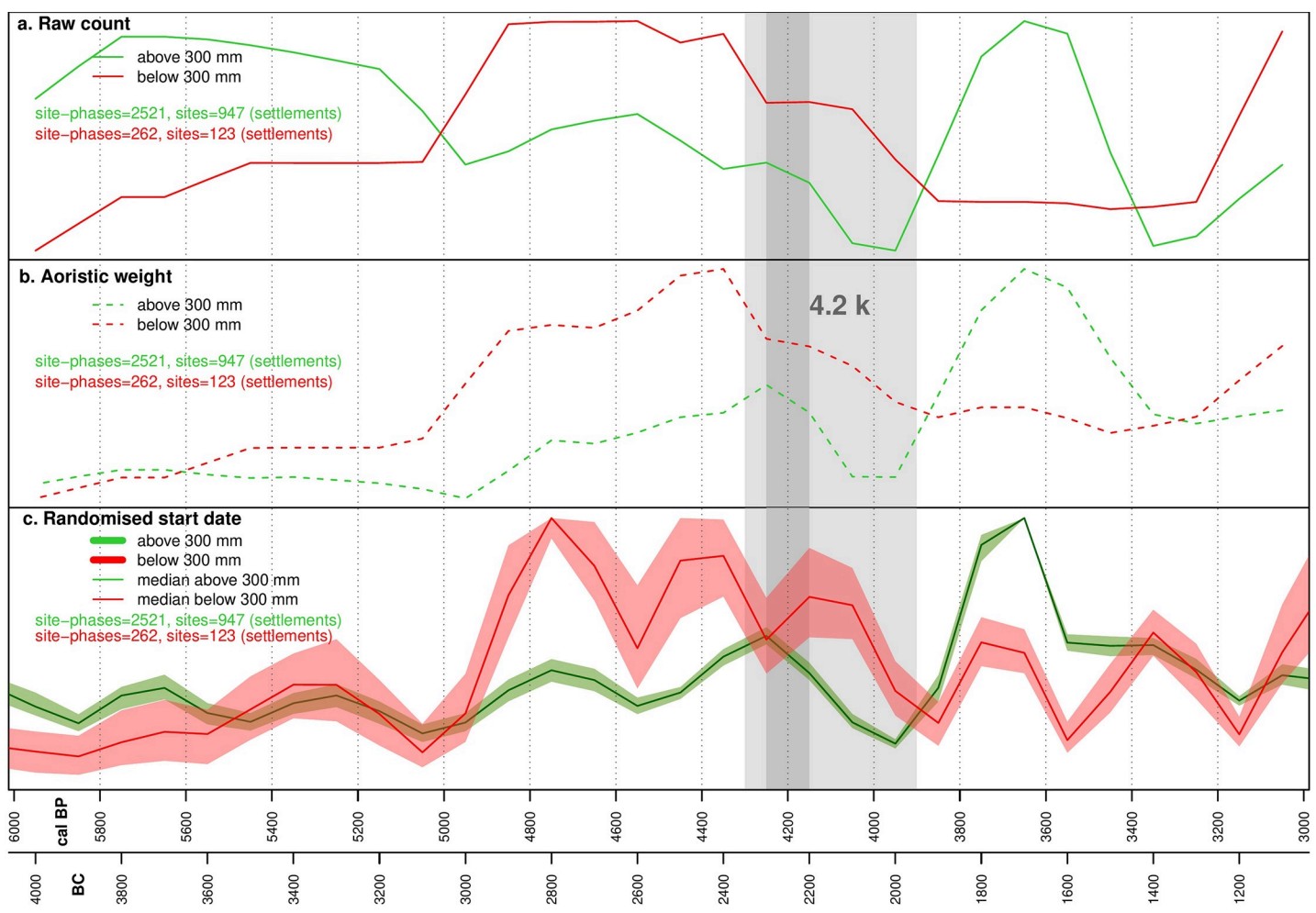

**Fig 4. Rural settlement dataset trends by rainfall.** a: Raw site counts. b: Aoristic weight. c: Randomized start dates.

and 1 for ease of comparison. The survey dataset exhibits four broad stages. Initially, between 6,000 and 5,000 cal BP, settlement is fairly stable. This is in part due to poorly refined chronologies in this period, with many sites simply labelled as Chalcolithic (5,400–3,000 cal BP), and this explains the marked difference between the raw counts (Fig 4A) and aoristic weights (Fig 4B) in the wetter zone. Where phase lengths are very long, raw count will tend to overestimate the population since successive occupations are lumped together. For this reason, we favor the aoristic weights as a proxy for population levels in this initial phase. After 5,000 cal BP, phase lengths decrease and the different models correlate well. Between 5,000 and 4,300 cal BP population in both zones increased, particularly in the dryer zone. Decline began at 4,300 cal BP in this region, and slightly later in the wetter zone, continuing through the 4.2kya event time window. The final phase saw a second peak in settlement in the wetter zone at around 3,600 cal BP, with no corresponding rise in the dryer zone (Fig 4). The urban dataset exhibits a broadly similar pattern to the surveys (Fig 5), with an initial phase of relatively low population from 6,000 to 5,000 cal BP concentrated in fertile areas above the 300 mm isohyet (Fig 6), followed by a rise and peak at around 4,200 cal BP. Decline during the period of the 4.2kya event is visible in the dryer zone but is much less significant in better watered areas in both raw site count (Fig 5A) and total area (Fig 5B), and absent in the aoristic weight curve (Fig 5C). This is in part

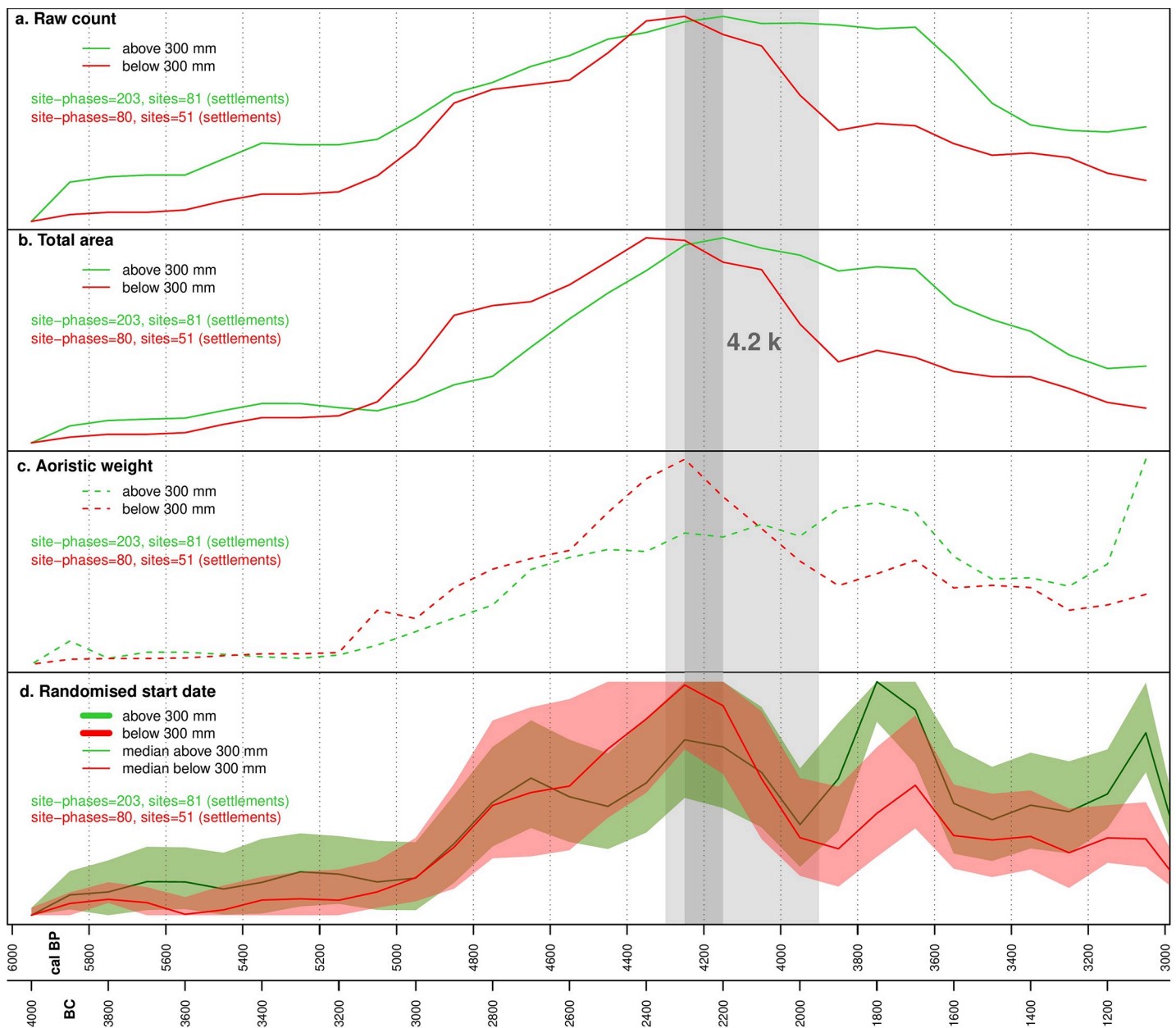

**Fig 5. Urban settlements dataset trends.** a: Raw site counts. b: Total area (summed). c: Aoristic weight (of counts). d: Randomized start dates (of counts).

due to the emergence of a new cluster of large urban sites in the west of the study region visible in the KDE plots (Fig 6). A more precipitous decline does occur across all urban sites but not until around 3,700 cal BP.

## Comparative results

We use Pearson's correlation coefficient to compare between the different datasets and modelled proxies. The raw count of surveyed sites is poorly correlated with the demographic trends inferred by aoristic weights and randomized start dates, while the latter are significantly correlated (Table 3). As said above, this is due to the fact that the raw count of sites tend to

# Urban Area 200 Year Snapshots

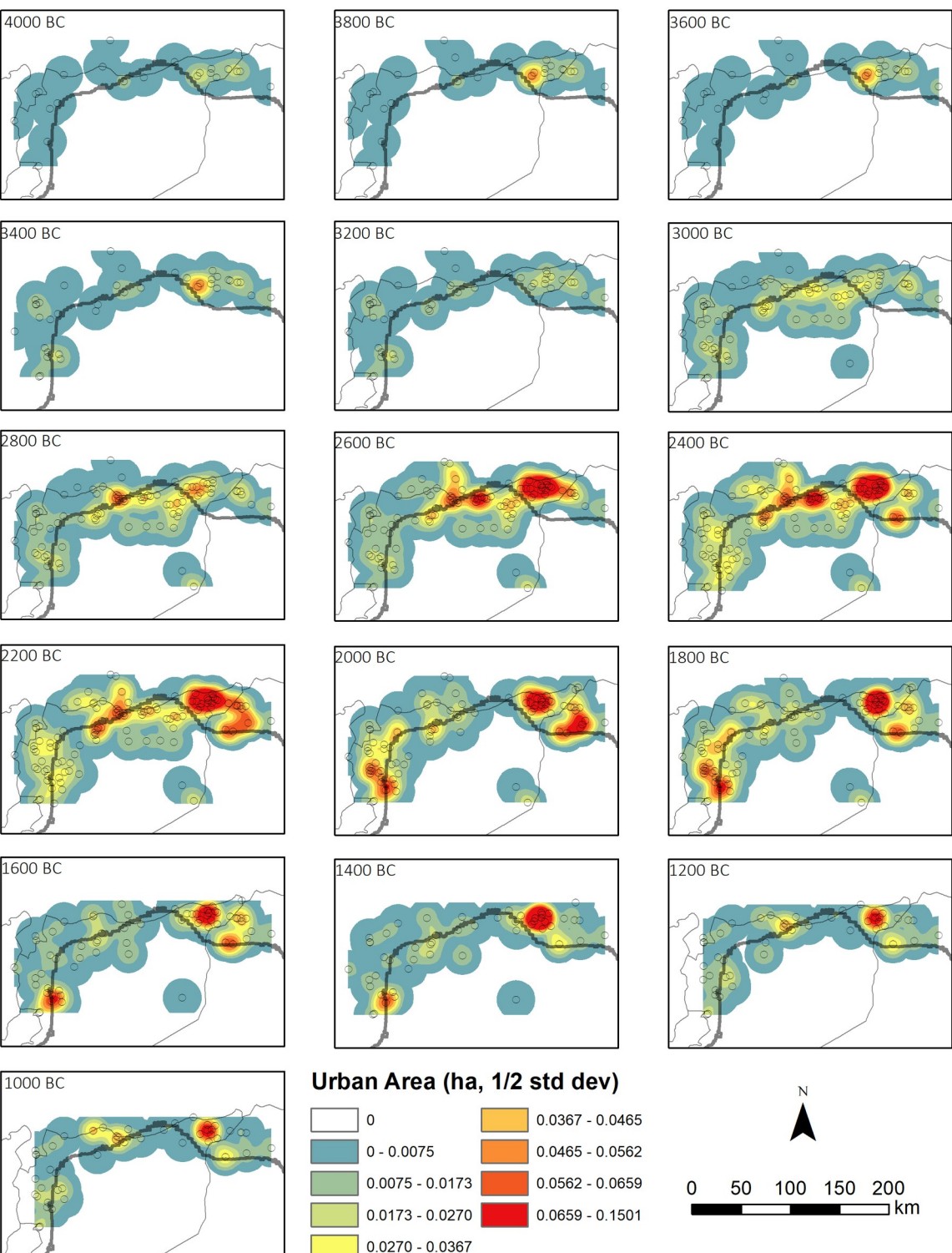

**Fig 6. Urban settlement kernel density estimates.** KDE bandwidth: 50km. Snap shot maps at 200-year intervals. Country outlines from the GADM website (https://gadm.org/).

**Table 3. Pearson correlation coefficient r-value matrix between the archaeo-demographic proxies (all sites) from 6,000 to 3,000 cal BP.**

|                   | Count | Aoristic weight | Random | SPD  |
|-------------------|-------|-----------------|--------|------|
| **Count**         | 1.00  |                 |        |      |
| **Aoristic weight** | 0.04  | 1.00            |        |      |
| **Random**        | 0.14  | **0.86**        | 1.00   |      |
| **SPD**           | -0.18 | 0.25            | 0.16   | 1.00 |

Significant correlations indicated by bold numbers (p-value < 0.05).

overestimate the population in the earlier periods as many unexcavated surveyed sites have been dated to a generic Late Chalcolithic phase. There is no correlation with the SPD dataset across the full range of the study period because the SPD does not capture demographic changes recorded in the survey data during the Middle Bronze Age (4,000–3,600 cal BP) and the Early Iron Age I (3,200–3,000 cal BP) (Fig 7). Comparing SPDs and survey dataset proxies

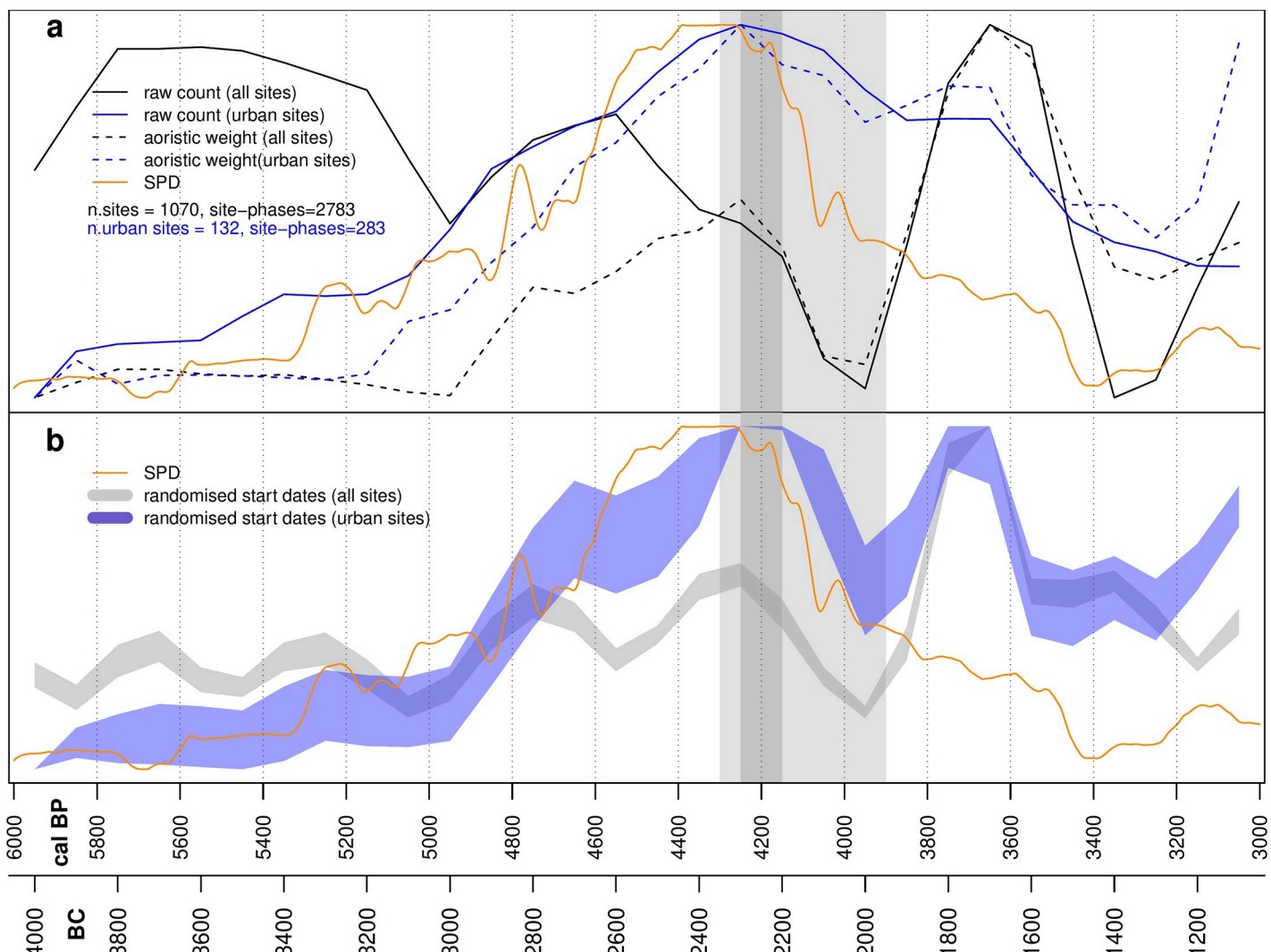

**Fig 7. Comparison of SPDs, urban and rural settlement.** a: SPD, raw counts and aoristic weights for urban and all sites. b: SPD and randomized start dates for urban and all sites.

**Table 4. Pearson correlation coefficient r-value matrix between the archaeo-demographic proxies (all sites) from 6,000 to 4,000 cal BP.**

|                  | Count | Aoristic weight | Random | SPD  |
|------------------|-------|-----------------|--------|------|
| **Count**        | 1.00  |                 |        |      |
| **Aoristic weight** | -0.39 | 1.00         |        |      |
| **Random**       | -0.26 | **0.87**        | 1.00   |      |
| **SPD**          | -0.61 | **0.91**        | **0.70** | 1.00 |

Significant correlations indicated by bold numbers (p-value < 0.05).

between 6,000 and 4,000 cal BP, rather than 3,000 cal BP, reveals a statistically significant correlation (Table 4).

The counts, areas, aoristic weights and randomized start date proxies for the urban dataset are all significantly correlated (Table 5) and present a similar picture (Fig 6). The SPD dataset is well correlated with all metrics derived from the urban dataset (Table 5). This may in part be due to sampling bias, since excavation projects in the study region, and particularly the sorts of large well-funded projects likely to provide large numbers of radiocarbon dates, tend to concentrate on larger sites. The population spikes after 4,000 cal BP present in the survey dataset are also less marked in the urban dataset (Fig 6). As with the survey dataset, all correlations between the SPD and urban dataset improve if the analysis is performed only from 6,000–4,000 cal BP (Table 6), suggesting the SPD is not capturing fluctuations in settlement during the period after 4,000 cal BP as well.

Although there is some variability in the trends in each of our datasets and the different probabilistic models applied, contextual analysis of the results does suggest a degree of correlation (Fig 7). Discounting the raw count for the rural sites (see above), all three datasets show a similar pattern up until 4,500 cal BP, with a gradual and then increasingly rapid rise in inferred population. After 4,500 cal BP, total rural settlement starts to decline while urban site numbers and the SPD continue upwards. Both the urban and SPD datasets peak between 4,400 and 4,200 cal BP, after which there is a rapid decline in the SPD over only around 100 years, followed by a slower paced decrease. Urban site counts also fall from 4,200 cal BP but this is much more gradual, and the overall numbers are fairly stable until the end of the Middle Bronze Age at 3,600 cal BP. This difference between the SPD and urban site datasets could be the result of differences in chronological precision, such that a short period of reduced activity is visible in the relatively chronologically sensitive SPD but is not evident in chronological attributions reliant on ceramic changes. However, if this were the case we might expect the SPD to recover into the Middle Bronze Age (4,000–3,600 cal BP) when urban site counts remain high and rural sites experience a second peak. The fact that the SPD does not capture this may be due to research biases towards excavation at Early Bronze Age sites and phases

**Table 5. Pearson correlation coefficient r-value matrix between the archaeo-demographic proxies (only urban sites) from 6,000 to 3,000 cal BP.**

|                  | Count   | Area    | Aoristic weight | Random  | SPD  |
|------------------|---------|---------|-----------------|---------|------|
| **Count**        | 1.00    |         |                 |         |      |
| **Area**         | **0.97**| 1.00    |                 |         |      |
| **Aoristic weight** | **0.88** | **0.91** | 1.00       |         |      |
| **Random**       | **0.88**| **0.89**| **0.96**        | 1.00    |      |
| **SPD**          | **0.85**| **0.78**| **0.66**        | **0.66**| 1.00 |

Significant correlations indicated by bold numbers (p-value < 0.05).

**Table 6. Pearson correlation coefficient r-value matrix between the archaeo-demographic proxies (only urban sites) from 6,000 to 4,000 cal BP.**

|  | Count | Area | Aoristic weight | Random | SPD |
|---|---|---|---|---|---|
| **Count** | 1.00 |  |  |  |  |
| **Area** | **0.98** | 1.00 |  |  |  |
| **Aoristic weight** | **0.98** | **0.99** | 1.00 |  |  |
| **Random** | **0.97** | **0.98** | **0.98** | 1.00 |  |
| **SPD** | **0.95** | **0.93** | **0.94** | **0.93** | 1.00 |

Significant correlations indicated by bold numbers (p-value < 0.05).

within sites, and the fact that several large Middle Bronze Age site were excavated before radio-carbon dating was routinely employed. All proxies indicate a period of relatively limited settlement during the Late Bronze Age (3,600–3,200 cal BP), while the beginning of a new period of dense rural settlement and urban expansion is visible from the beginning of the Iron Age (3,200 cal BP onwards) in the urban and rural site datasets. The SPD does not capture this shift, likely due to fewer excavations being conducted on sites dating to this period, and because the excavators routinely rely on material culture rather than radiocarbon for dating.

## Discussion

Taken at face value, our datasets indicate support for claims of a decline in population and social complexity contemporary with the 4.2kya event, with negative trends in all three datasets between 4,300 and 3,900 cal BP and particularly sharp drops in the SPD and rural settlement proxies. However, the temporal and spatial scale of our study allows us to add nuance to a straightforward model of climate deterioration causing the collapse of an otherwise stable system. Over the longer term we can see a cycle of boom and bust in urbanization, with the boom beginning at around 5,000 cal BP and increasing significantly from 4,800 cal BP before the decline from around 4,200 cal BP. This is visible in both the urban sites dataset and the SPD. Urban sites are founded and expand across the landscape but the drier zone experiences the greatest growth in both urban site area (Fig 6) and rural site counts, and the greatest subsequent decline (see Figs 3–5). We have previously argued [36,37] that a new focus on textile production was a primary driver of both urbanization and settlement expansion into these drier regions. At the major urban site of Ebla, textual evidence suggests the palace directly controlled flocks of sheep and goat numbering hundreds of thousands of animals [73] and was also supplied with wool from surrounding regions [74]. Sheep and goat could be pastured in more marginal environments, while the shift from flax to wool as the source of material for textile production would also have freed up prime agricultural land for food production [75]. The urbanization and population growth seen across the study region was likely underpinned by an expansion in productive capacity, leading to an economic boom by the middle of the third millennium BC. The expansion into more arid environments increased the risk of crop failure. Urban-based institutions, such as the palace at Ebla, may have operated to buffer the uncertainty created by the risks involved. However, the ability to manage risk in this way may also have encouraged risk-taking behavior [76] and rendered the system more reliant on elite management and contingent political configurations. Higher stocking rates may have led to ecological degradation and desertification associated with overgrazing. Desertification as a result of overgrazing has been recorded in dryland environments across the world [77 and references therein], including in the modern Levant [78] and in Late Chalcolithic and Early Bronze Age Jordan, again linked to overstocking for commercialized textile production [79]. Within our study area, land use reconstructions based on archaeobotanical data have shown

an increase in desert-steppe environments in the Jazira during the Middle Bronze Age (4,000–3,800 cal BP) when compared to the Early Bronze Age (5,000–4,000 cal BP), attributed to ecological degradation caused by both overgrazing and agricultural intensification [80]. The absence of sedentary settlement and information from historical sources such as the Mari letters suggests that across much of the drier parts of our study area the Middle Bronze Age saw a shift in land-use towards mobile pastoralism [17,81].

From 4,600 to 4,100 cal BP the SPD exceeded the envelope of the logistic null model. The model assumes that population growth slows over time as resources become scarce and the carrying capacity of the environment is reached. Carrying capacity refers to the maximum population which can be sustainably supported by a particular environment. Exceeding the range predicted by the logistic model therefore indicates a population increase beyond the long-term sustainability of the environment in a condition known as overshoot [82]. This would suggest that the 4.2kya event occurred during a period of unprecedented urban and rural growth which may have been unsustainable even without an exogenous climate event. This does not preclude the event acting as a proximate cause for decline; without it, the system may have stayed in an overshoot phase for a longer period before returning to sustainable levels. Furthermore, the logistic model used here assumes carrying capacity is fixed, and therefore does not take into account increasing productivity brought about through technological change or, importantly in this case, the expansion of production into previously underexploited areas such as the drier zone. Put another way, however, the model suggests that increased population and urbanization was supported either through unsustainable overshoot, or through the exploitation of a drier zone with increased risk of crop failure. We would argue that the expansion of settlement and population, alongside the high density of urban centers, resulted in a series of socio-environmental mismatches [83], reducing the sustainability of the system as whole. The overall trends presented above also mask substantial variability. During the period of maximum urban growth from 4,600 cal BP, the Eastern Jazira attained a density of urban sites which was not matched until the 20[th] century [84], while the population levels reached in the drier zone were not repeated until the later part of the Iron Age when new water management technologies transformed agricultural potential [47]. Studies of individual urban centers have found that their local environment could not meet their resource requirements in a sustainable way [85]. However, areas such as Western Syria show continuity of settlement, and even substantial urban growth, over the 4.2kya time window.

## Conclusions

Debates on the collapse of past societies often suffer from 'terminological ambiguity' [86]; the speed, magnitude and nature of change required to count as 'collapse' is left undefined [67]. Narratives of collapse around the 4.2kya event in the Middle East have tended to emphasize a political collapse, of the Akkadian Empire, a socio-economic collapse, indicated by declines in the size of urban centers, and a population collapse, indicated by the abandonment of sites. The datasets we have presented here do not provide sufficient temporal precision or cultural attribution to comment on the question of the Akkadian Empire itself. Unlike in earlier periods, such as the Uruk, or later territorial empires, archaeologists cannot currently recognize clear ceramic indicators directly and unambiguously associated with Akkadian presence [87]. This means we cannot detect when or under what conditions Akkadian control in Upper Mesopotamia came to an end. However, our datasets do allow us to identify trends in urban site trajectories and population. Two of our three proxies, the SPD and rural site counts, show rapid and marked declines during the 4.2kya window. Urban site sizes and counts also declined but more gradually. However, by changing the spatial and temporal scale of analysis

we can see that the specific socio-economic system in place during the period of the 4.2kya event was more vulnerable than in earlier periods. In this state even a small exogenous change may have impacted the overall system, but it is also possible that the shift away from urbanism may have been the result of endogenous developments. More broadly, we can see a millennial scale pattern of urban emergence and decline between 5,000 and 3,000 cal BP, and regional variation in responses to the 4.2kya event. In some parts of our study area, particularly in more well-watered areas such as Western Syria, the event is barely visible at all and local dynamics are more significant [88]. Future research on what enabled communities at some sites, and even regions, to survive and others to fail is needed, but must be undertaken at appropriate spatial and temporal scales [89].

## Supporting information

**S1 Data.**
(XLSX)

## Acknowledgments

We are grateful to three reviewers for insightful comments which improved the quality of the paper.

## Author Contributions

**Conceptualization:** Dan Lawrence.

**Data curation:** Alessio Palmisano, Michelle W. de Gruchy.

**Formal analysis:** Alessio Palmisano, Michelle W. de Gruchy.

**Funding acquisition:** Dan Lawrence.

**Writing – original draft:** Dan Lawrence.

**Writing – review & editing:** Alessio Palmisano, Michelle W. de Gruchy.

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
