## [Decision Letter · Decision Letter 0]

28 Oct 2020

PONE-D-20-30893

Collapse and Continuity: A multi-proxy reconstruction of settlement organization and population trajectories in the Northern Fertile Crescent during the 4.2kya Rapid Climate Change event

PLOS ONE

Dear Dr. Lawrence,

Thank you for submitting your manuscript to PLOS ONE. After careful consideration, we feel that it has merit but does not fully meet PLOS ONE’s publication criteria as it currently stands. Therefore, we invite you to submit a revised version of the manuscript that addresses the points raised during the review process.

All comments need to be addressed before re-submission.

We look forward to receiving your revised manuscript.

Kind regards,

Peter F. Biehl, PhD

Academic Editor

PLOS ONE

Journal Requirements:

2. We note that Figures 1, 2 and 6 in your submission contain map/satellite images which may be copyrighted.

a. You may seek permission from the original copyright holder of Figures 1, 2 and  to publish the content specifically under the CC BY 4.0 license. 

Additional Editor Comments:

Your manuscript has now been seen by two referees, whose comments are appended below. You will see from these comments that while one of the referees finds your work of potential interest, the other other reviewer has raised substantial concerns that must be addressed. In light of these comments, we cannot accept the manuscript for publication, but would be interested in considering a revised version that addresses these serious concerns.

We hope you will find the referees' comments useful as you decide how to proceed. Should presentation of further data and analysis allow you to address these criticisms, we would be happy to look at a substantially revised manuscript. However, please bear in mind that we will be reluctant to approach the referees again in the absence of major revisions.

Reviewers' comments:

Reviewer's Responses to Questions

**Comments to the Author**

1. Is the manuscript technically sound, and do the data support the conclusions?

Reviewer #1: No

Reviewer #2: Yes

2. Has the statistical analysis been performed appropriately and rigorously? 

Reviewer #1: No

Reviewer #2: Yes

3. Have the authors made all data underlying the findings in their manuscript fully available?

Reviewer #1: No

Reviewer #2: Yes

4. Is the manuscript presented in an intelligible fashion and written in standard English?

Reviewer #1: Yes

Reviewer #2: Yes

5. Review Comments to the Author

Reviewer #1: 1. Blurs high resolution 14C chronology in multiple regions into arbitrary 100 year bins, when the events and processes described are measurably visible at AMS-calibrated Oxcal-regulated and Bayes- manipulated decadal levels;

2. Misses Spain, France, Italy, Greece, Palestine, Egypt, Indus and East Asia societal collapses synchronous at 4.2 ka BP with northern Levant and northern Mesopotamia ("northern Fertile Crescent");

3. Misses westerlies, ISM, and EASM synchronous, high resolution, abrupt 4.2 ka BP event;

4. Misses high resolution northern Levant and north Mesopotamian region proxy data for abrupt, century-scale, 4.2 ka BP event;

5. Takes inexpert, pop archaeology, opinions (e.g., Middleton), lacking any data, for facts of Akkadian collapse;

6. Guesses at testing of regional societies' carrying capacity limit at end 3rd mill., when post 4.2 ka BP event settlement and population were both successful and multiple times greater across this region;

7. Takes SPDs as representative sample of human activity across time and space, and mysteriously of population, when SPDs only reflect archaeologists’ skewed 14C sampling, if any, across time and space (Palmisano et al 2017 lists all the errors in using SPDs), that is, "a hopeless endeavor" (Torfing 2015).

Illustrates how high resolution archaeological and paleoclimate data can be reconstructed into low resolution data, and thereby deployed to ambiguously support an otherwise disproven thesis.

Reviewer #2: The approach presented in this manuscript is a welcome addition in archaeology, especially the kind of research that combines long-term climatic changes and regional settlement patterns. Although the research area encompasses several different climate zones and the approach may distort valuable sub-regional variations in climatic and cultural spheres, we need to test supra-regional approaches and analyses of archaeological data. I have to stress however, that 4.2 is far from being agreed upon as a global event as authors state in line 76!

6. PLOS authors have the option to publish the peer review history of their article (what does this mean?). If published, this will include your full peer review and any attached files.

Reviewer #1: No

Reviewer #2: No

---

## [Author Response · Author response to Decision Letter 0]

2 Nov 2020

Reviewer 1

1. Blurs high resolution 14C chronology in multiple regions into arbitrary 100 year bins, when the events and processes described are measurably visible at AMS-calibrated Oxcal-regulated and Bayes- manipulated decadal levels;

The technique used and described in the paper does not blur chronological dates and does not assign individual dates to 100 year bins. We use an established technique (see citations in the paper) to mitigate the impact of single site phases with multiple radiocarbon dates close together on our curves. This uses 50 year bins (not 100 year bins) and is only applied to sites where there are multiple C14 dates within 50 years of one another (as explained in lines 190-192). Where we have multiple dates within 50 years the uncalibrated radiocarbon age and error of all dates are averaged, and we then calibrate the averaged date rather than multiple dates. We want to stress that this technique is only applied to a small number of phases at a small number of sites where there are multiple dates within 50 years. All other dates contribute to the curves at the resolution of the published date, and the graphs in the relevant figures are at a yearly resolution. We have added text clarifying this at line 194, in addition to the explanation at 189-193 which covers the above points. 

Although not explicitly stated, we understand the reviewer to be concerned that the binning technique we use will mask very short-term fluctuations in the curve which can be attributed to the effects of the 4.2k event. We hope that the fact that we are applying a 50 year rather than 100 year bin will mitigate this concern, along with the clarification that we are only applying this to a limited number of sites. More fundamentally, the effect on the dataset is not to blur or coarsen it, but to smooth fluctuations resulting from sampling biases (something which the reviewer raises as important in point 7 below). We would also note that at a regional scale the duration of the 4.2k event is considered to be centennial rather than decadal (as we state in lines 79-81, with citations), meaning the resolution of our C14 dataset should be sufficiently precise to identify fluctuations resulting from it. In fact, as we state in lines 348-350 and show in Figures 2 and 7, we can see a rapid decline at 4.2k over only 100 years. We argue in lines 351-355 that it is the increased precision of C14 dates relative to other proxies which allows us to ‘see’ the event in this dataset. 

2. Misses Spain, France, Italy, Greece, Palestine, Egypt, Indus and East Asia societal collapses synchronous at 4.2 ka BP with northern Levant and northern Mesopotamia ("northern Fertile Crescent");

Our paper does not pretend to include all of the regions where the 4.2kya event has been considered to have an impact, and we state the location of the study region in the title, abstract and introduction. We also acknowledge that the event is considered to have impacted societies beyond out study region (lines 55-56, 85-88) and provide several citations to support this. As we state in lines 85-92, the goal of our study is to investigate the event at a regional scale using empirical data as opposed to synthetic comparison, and the area we have chosen is based on a shared physical geography. A future project might compare all of the regions described by the reviewer using archaeological data but this would require years if not decades of data collection. While this work is undertaken, we feel regional analyses can still advance the debate on this topic, as Reviewer 2 suggests. 

3. Misses westerlies, ISM, and EASM synchronous, high resolution, abrupt 4.2 ka BP event;

As stated in lines 66-68, the goal of this paper is to examine population and settlement organisation during the period of the 4.2kya event. We take it as fact that the event occurred in some form, and briefly explain the state of the debate on its magnitude and duration with appropriate citations (lines 71-82). As such we do not go into detail on the climate forcing mechanisms, such as the westerlies, Indian Summer Monsoon (ISM) and East Asian Summer Monsoon (EASM) shifts mentioned by the reviewer, which may account for the climate event itself. We are aware of these mechanisms, and cite relevant literature which deals with them, including our own work (Cookson et al. 2019), but we do not feel that further discussion is relevant to the goals of the paper. We have added text clarifying that we are not dealing with the climate proxies here in lines 82-84 and an additional date range in line 80.

4. Misses high resolution northern Levant and north Mesopotamian region proxy data for abrupt, century-scale, 4.2 ka BP event;

We are unsure what the reviewer is referring to here, but assume they mean climate proxy data. If this is the case then as above we stress that our paper is not addressing the climate proxy data, although we do cite several overview papers on climate proxy data derived from natural archives (speleotherms, pollen, formanifera etc) in the region. We have added text clarifying this in lines 82-84 and an additional date range in line 80. If the reviewer is referring to specific proxy data for another variable then we would be happy to address this concern if provided with more detail. 

5. Takes inexpert, pop archaeology, opinions (e.g., Middleton), lacking any data, for facts of Akkadian collapse;

We cite multiple sources on the collapse of the Akkadian Empire, all of which come from peer reviewed journal articles, books and edited volumes. None of these would be considered popular archaeology. The two citations of Middleton are from a Cambridge University Press single author volume and a chapter in an edited volume. The CUP volume is not a popular science volume and was (favourably) reviewed in several academic publications including the European Journal of Archaeology (Smart 2018) and the American Journal of Archaeology (Tainter 2018) as an academic work. The chapter cited comes from a volume edited by Eustathios Chiotis which is also not a popular science book. All contributions to this volume were peer reviewed and the authors are all established academics. As such we feel this comment is unwarranted. We also note that the reviewer does not identify any substantive criticisms of our treatment of the Akkadian Empire in the paper. Without knowing what facts are incorrect we cannot amend the manuscript.

6. Guesses at testing of regional societies' carrying capacity limit at end 3rd mill., when post 4.2 ka BP event settlement and population were both successful and multiple times greater across this region;

The carrying capacity limit is not a guess, it is a function of the logistic model for population growth we use as a null model. The model assumes that the long term growth rate of pre-industrial populations fits a logistic curve. The logistic population curve assumes population gradually increased before reaching a limit based on the carrying capacity of the land. Deviations above the modelled 95% confidence envelope represent periods of population growth in which carrying capacity is exceeded. This model has been used in a number of papers (cited in our paper) to examine population fluctuations using SPDs. We have added text at lines 214-215 and 219-222 to clarify this. We already note the limitations of the logistic model in lines 398-401. 

The claim by the reviewer that ‘post 4.2 ka BP event settlement and population were both successful and multiple times greater across this region’ is not relevant in assessing the results of the logistic model, since these could also be phases of overshoot. More importantly, this claim is not supported by the data. Some individual surveys, notably around Tell Leilan, do experience substantial increases in settlement during the Middle Bronze Age, but at a regional scale rural settlement densities are comparable to those of the pre-4.2k phase and urban sites decline in density (see Figure 7 in our paper). Elsewhere (Lawrence et al. 2016, PLOS, Lawrence et al. 2017, Quat Int, Palmisano et al., in press, Quat Sci Rev) we have shown that population remains relatively low until the middle of the Iron Age in the Eastern part of our study area, and even later in the Western part. By this point very different technological and socio-political formations are in play. 

7. Takes SPDs as representative sample of human activity across time and space, and mysteriously of population, when SPDs only reflect archaeologists’ skewed 14C sampling, if any, across time and space (Palmisano et al 2017 lists all the errors in using SPDs), that is, "a hopeless endeavor" (Torfing 2015).

Palmisano is one of the authors of our paper. His 2017 paper cited by the reviewer does outline several criticisms of the SPD approach, but argues that SPDs of C14 dates are still a useful proxy when appropriately contextualised and compared with other proxies for population. This is precisely what we do in our paper. We acknowledge the limitations of the SPD approach in lines 103-112, including all criticisms outlines by Palmisano 2017. Here we also cite several recent papers both within and adjacent to our study region which find a good correlation between SPDs and other population proxies derived from settlement data. In the results section we use Pearson’s correlation coefficient to compare between the SPD and settlement derived proxies. We describe under what conditions we find statistically robust correlations between the different proxies and show that for the period between 6,000 and 4,000 the correlations between settlement derived proxies and SPDs are robust. In the discussion section we contextualise the relationship between SPDs and settlement datasets (e.g. lines 370-371, 376-377). 

Overall we feel that SPDs, although, like all proxies (including landscape survey data), imperfect, can provide insights into past population if handled appropriately. We also feel that a blanket assertion that ‘SPDs only reflect archaeologists’ skewed 14C sampling’ is not sustainable. We can support this claim through statistical examination of the relationship between SPD and landscape survey derived population curves, both within our paper and others cited in it covering parts of our study region and adjacent areas. More generally, SPDs have been widely used for two decades in several regions of the world, including in several papers published by PLOS. 

We note that the quotation from Torfing 2015 is from a paper on the use of SPDs in European prehistory. The full quote reads: ‘For the Mesolithic–Neolithic transition and the Neolithic in general, this seems, to the present author, a hopeless endeavour’. Since we are dealing with a different period and region we do not feel this criticism is relevant here. 

Illustrates how high resolution archaeological and paleoclimate data can be reconstructed into low resolution data, and thereby deployed to ambiguously support an otherwise disproven thesis.

As explained above (response to points 3 and 4), we do not use paleoclimate data in this paper and have added text clarifying this in lines 82-84. We also do not reconstruct high resolution data into low resolution data. In fact, the modelling techniques we have applied to the settlement data provide modelled values at an increased resolution. As explained above (response to point 1), the binning technique applied to the radiocarbon dates does not impact the resolution of the dataset in the manner suggested by the reviewer and we have added text clarifying this at line 194.

Reviewer 2

1. The approach presented in this manuscript is a welcome addition in archaeology, especially the kind of research that combines long-term climatic changes and regional settlement patterns. Although the research area encompasses several different climate zones and the approach may distort valuable sub-regional variations in climatic and cultural spheres, we need to test supra-regional approaches and analyses of archaeological data. 

We acknowledge that further investigation of sub-regional trends would be extremely interesting (and in fact have published longer term trends by region in PLOS, Lawrence et al. 2016) and we acknowledge this in the paper (e.g. lines 406-413, 432-433). However, given the volume of data involved we feel that detailed discussion of regional trends would be too much for this paper. 

2. I have to stress however, that 4.2 is far from being agreed upon as a global event as authors state in line 76!

We agree with this statement but wanted to convey the fact that the International Commission on Stratigraphy have recently established the 4.2k event as a boundary marker for the end of the Middle and beginning of the Late Holocene, suggesting consensus in at least one field. We have amended the sentence to clarify this point (lines 76-78).

---

## [Decision Letter · Decision Letter 1]

9 Dec 2020

PONE-D-20-30893R1

Collapse and Continuity: A multi-proxy reconstruction of settlement organization and population trajectories in the Northern Fertile Crescent during the 4.2kya Rapid Climate Change event

PLOS ONE

Dear Dr. Lawrence,

Thank you for submitting your manuscript to PLOS ONE. After careful consideration, we feel that it has merit but does not fully meet PLOS ONE’s publication criteria as it currently stands. Therefore, we invite you to submit a revised version of the manuscript that addresses the points raised during the review process.

Though most of the comments have been addressed there are remaining issues indicated by an additional reviewer (reviewer 3) which need to be addressed before acceptance.

We look forward to receiving your revised manuscript.

Kind regards,

Peter F. Biehl, PhD

Academic Editor

PLOS ONE

Additional Editor Comments (if provided):

Though most of the comments have been addressed there are remaining issues indicated by an additional reviewer (reviewer 3) which need to be addressed before acceptance.

Reviewers' comments:

Reviewer's Responses to Questions

**Comments to the Author**

1. If the authors have adequately addressed your comments raised in a previous round of review and you feel that this manuscript is now acceptable for publication, you may indicate that here to bypass the “Comments to the Author” section, enter your conflict of interest statement in the “Confidential to Editor” section, and submit your "Accept" recommendation.

Reviewer #1: (No Response)

Reviewer #2: All comments have been addressed

Reviewer #3: All comments have been addressed

2. Is the manuscript technically sound, and do the data support the conclusions?

Reviewer #1: No

Reviewer #2: Yes

Reviewer #3: Yes

3. Has the statistical analysis been performed appropriately and rigorously? 

Reviewer #1: No

Reviewer #2: Yes

Reviewer #3: Yes

4. Have the authors made all data underlying the findings in their manuscript fully available?

Reviewer #1: No

Reviewer #2: Yes

Reviewer #3: Yes

5. Is the manuscript presented in an intelligible fashion and written in standard English?

Reviewer #1: Yes

Reviewer #2: Yes

Reviewer #3: Yes

6. Review Comments to the Author

Reviewer #1: (No Response)

Reviewer #2: My issues have been addressed. I believe that the regional focus of the MS and the level of detail in discussions are sufficient for PLOS One.

Reviewer #3: The manuscript by Lawrence et al is a very interesting and well written review of archaeological data and radiocarbon dating results aimed at investigate the role of the 4.2 BP event in the so-called collapse of the Akkadian Empire. Since Weiss et al proposed a deterministic, climate-triggered collapse of the Akkadian Empire, this is one of the most relabel and robust attempt to propose an alternative hypothesis to explain archaeological data. And, I think, authors convincingly accomplish this task. As you can understand I am not a fan of deterministic interpretation of climatic/environmental changes as the major player in society collapse as much as I am not a fan of the idea of the concept itself of societal collapse. Also, I think that the 4.2 BP event was not global android not effected many areas of the world. As a geoarchaeologist, I use to work on solid data and I try to avoid general oversimplifications that early led to climatic determinism.

In a recent paper, I wrote this sentence: 'Emerging research has moved beyond simplistic and linear interpretations of antiquity, invoking anthropological paradigms of continuity, social resilience and transformation, as well as new methodological approaches for resolving how cultures may have assimilated, or coped by strategic adaptation, migration, socio-political reorganization or technological innovation' (Nicoll and Zerboni, Quat Int 2020). I think that the manuscript by Lawrence et al perfectly reflect this sentence, because looks at the issue of the collapse of the Akkadian Empire considering data and not following the mainstream. Moreover, I considered the comments and suggestions of previous reviewers and authors reply (that are very good) and thus I strongly suggest to accept the manuscript for publication pending very minor revisions.

I am not expert in spatial analyses and statistics applied to a archaeological data, but I work since several years with researched focused on these topics and I think I can confirm that the methods here proposed is reliable as much as the dataset considered, which includes data from a number of recent field surveys carried out in a region spanning from the Mediterranean shorelines of Syria to the Zagros Mt. Also, such field survey were carried out by major research groups and their result are highly valuable. The comparison of archaeological and demographic data with ecological concepts is the winning idea at the basis of this paper. Maybe some more sentences at the beginning go the discussion explaining the concept of carrying capacity of the environment would help the reader to understand the conclusion of the discussion. I am deeply convinced that the explanation here offered of a crisis of the economic system and settlement pattern occurred after the concomitance of increased aridity and out-of-control demographic growth is the most reliable. In my experience in N Italy, we used the same concepts to explain the so-called collapse of the Terramare (Bronze Age) culture, which happened after a period of demographic growth and deterioration of the soil triggered by aridification and overgrazing (Cremaschi et al., 2016 Quat. Sc. Rev.).

The story this paper tells of continuity and adaptation; in fact, I would like to understand whether this is a case of population relocation and/or major change in land use. As you are considering the hinterlands of the main Akkadian capital (which is the crucial area where to investigate social instability and collapse), can you detect in this phase a shift from agricultural to pastoral vocation of villages? Do you have evidence to establish whether a shift in land use as a tuning factor or a consequence of the social reorganisation?

In the discussion you discuss the decrease in productivity related to increased exploitation of natural resources. Also this is a major point that can be explained more in details. There is a number of modern and paleo case studies illustrating how overgrazing under unstable climatic conditions lead to the disruption of ecological equilibrium and thus to the disruption of non-flexible land use strategies. You can find some example cited in Zerboni and Nicoll 2019 Geomorphology.

Finally, In the conclusion, I would underline the concept of continuity, which is the major take-home message of the manuscript.

Once a few sentences on these topics will be added, the paper will be ready for publication.

*** Please consider that I cited some of my recent papers on this topic to support your interpretation, no need to cite them!!!!!

7. PLOS authors have the option to publish the peer review history of their article (what does this mean?). If published, this will include your full peer review and any attached files.

Reviewer #1: No

Reviewer #2: No

Reviewer #3: **Yes: **Andrea Zerboni

---

## [Author Response · Author response to Decision Letter 1]

14 Dec 2020

'Maybe some more sentences at the beginning of the discussion explaining the concept of carrying capacity of the environment would help the reader to understand the conclusion of the discussion'

We have added sentences explaining the concept of carrying capacity and its relationship with the logistic growth model at lines 401-406. We have also added a reference to a similar use of this method.

'The story this paper tells of continuity and adaptation; in fact, I would like to understand whether this is a case of population relocation and/or major change in land use. As you are considering the hinterlands of the main Akkadian capital (which is the crucial area where to investigate social instability and collapse), can you detect in this phase a shift from agricultural to pastoral vocation of villages? Do you have evidence to establish whether a shift in land use as a tuning factor or a consequence of the social reorganisation?'

This is a very interesting point, and one that has not been addressed in detail by our paper or others. There is also a complicating factor here in that the phase immediately after the 4.2k event is when true nomadic pastoralism likely emerged in our study region. Because this form of existence leaves very limited material traces in the archaeological record, it is difficult to distinguish between population relocation and a land-use shift to mobile pastoralism – both result in an absence of evidence for sites in the landscape. In our current ERC project we are hoping to address this by gathering the limited archaeobotanical and zooarchaeological data available from excavations in the region to see if changes in animal exploitation are visible in the periods before, during and after the 4.2kya event. We will also use this so reconstruct land use through Species Distribution Modelling. However, we do not have these data as yet. For this paper we make reference to land use change in a new section in the discussion (lines 389-400) and discuss the limited evidence currently available for this (including new references). 

'In the discussion you discuss the decrease in productivity related to increased exploitation of natural resources. Also this is a major point that can be explained more in details. There is a number of modern and paleo case studies illustrating how overgrazing under unstable climatic conditions lead to the disruption of ecological equilibrium and thus to the disruption of non-flexible land use strategies. You can find some example cited in Zerboni and Nicoll 2019 Geomorphology.'

We are very grateful to Professor Zerboni for pointing us to this literature. We have added a section in the discussion (lines 389-400) which deals with overgrazing and the (limited) evidence available for this in our region. This includes evidence from our preliminary work on land use reconstruction and several other new references. As mentioned above, overexploitation, extensification and intensification are topics we intend to address in future papers once the data collection phase of our current project provides sufficient evidence to make firm statements on this.

'Finally, In the conclusion, I would underline the concept of continuity, which is the major take-home message of the manuscript.'

We have added a sentence emphasising this to the conclusion (lines 445-447) and an additional reference on continuity of occupation in Western Syria in particular.

---

## [Editor Report · Decision Letter 2]

18 Dec 2020

Collapse and Continuity: A multi-proxy reconstruction of settlement organization and population trajectories in the Northern Fertile Crescent during the 4.2kya Rapid Climate Change event

PONE-D-20-30893R2

Dear Dr. Lawrence,

We’re pleased to inform you that your manuscript has been judged scientifically suitable for publication and will be formally accepted for publication once it meets all outstanding technical requirements.

Kind regards,

Peter F. Biehl, PhD

Academic Editor

PLOS ONE
---

## [Editor Report · Acceptance letter]

29 Dec 2020

PONE-D-20-30893R2 

Collapse and Continuity: A multi-proxy reconstruction of settlement organization and population trajectories in the Northern Fertile Crescent during the 4.2kya Rapid Climate Change event 

Dear Dr. Lawrence:

I'm pleased to inform you that your manuscript has been deemed suitable for publication in PLOS ONE. Congratulations! Your manuscript is now with our production department. 

Kind regards, 

on behalf of

Dr. Peter F. Biehl 

Academic Editor

PLOS ONE